# Diverse functions of homologous actin isoforms are defined by their nucleotide, rather than their amino acid sequence

Pavan Vedula[1†], Satoshi Kurosaka[1†], Nicolae Adrian Leu[1], Yuri I Wolf[2], Svetlana A Shabalina[2], Junling Wang[1], Stephanie Sterling[1], Dawei W Dong[1,3], Anna Kashina[1]*

[1]Department of Biomedical Sciences, University of Pennsylvania, Philadelphia, United States; [2]National Center for Biotechnology Information, National Institutes of Health, Bethesda, United States; [3]Institute for Biomedical Informatics, Perelman School of Medicine, University of Pennsylvania, Philadelphia, United States

**Abstract** β- and γ-cytoplasmic actin are nearly indistinguishable in their amino acid sequence, but are encoded by different genes that play non-redundant biological roles. The key determinants that drive their functional distinction are unknown. Here, we tested the hypothesis that β- and γ-actin functions are defined by their nucleotide, rather than their amino acid sequence, using targeted editing of the mouse genome. Although previous studies have shown that disruption of β-actin gene critically impacts cell migration and mouse embryogenesis, we demonstrate here that generation of a mouse lacking β-actin protein by editing β-actin gene to encode γ-actin protein, and vice versa, does not affect cell migration and/or organism survival. Our data suggest that the essential in vivo function of β-actin is provided by the gene sequence independent of the encoded protein isoform. We propose that this regulation constitutes a global 'silent code' mechanism that controls the functional diversity of protein isoforms.
DOI: https://doi.org/10.7554/eLife.31661.001

*For correspondence:
akashina@upenn.edu

†These authors contributed equally to this work

Competing interests: The authors declare that no competing interests exist.

## Introduction

Actin is an essential and abundant intracellular protein that plays a major role in developmental morphogenesis, muscle contraction, cell migration, and cellular homeostasis. Two of the closest related actin isoforms, non-muscle β-actin and γ-actin, are ubiquitously expressed but encoded by different genes, producing nearly identical proteins except for four residues within their N-termini (*Vandekerckhove and Weber, 1978*). Notably, β-and γ-actin mRNA coding sequences differ much more significantly, by nearly 13%, due to silent substitutions affecting approximately 40% of their codons (*Erba et al., 1986*). Recent evidence suggests that this difference in mRNA coding sequence affects translation dynamics of the two actin isoforms: β-actin is translated in bursts and accumulates faster than γ-actin (*Buxbaum et al., 2014*; *Zhang et al., 2010*). Such faster translation results in their differential post-translational modification by arginylation, which targets only β- but not γ-actin (*Zhang et al., 2010*). Thus, actin isoforms are differentially regulated via changes in their mRNA coding sequence that can affect their translation and post-translationally modified state.

Despite their high similarity and abundance in the same cell types, β- and γ-actin play distinct non-redundant biological roles. A body of evidence shows that these actins localize to different parts of the cell and tend to incorporate into different actin cytoskeletal structures (*Dugina et al., 2009*; *Kashina, 2006*; *Otey et al., 1986*). More definitively, several studies show that β-actin knockout in mice results in early embryonic lethality despite proportional up-regulation of other actin isoforms to compensate for the total actin dosage (*Bunnell et al., 2011*; *Shawlot et al., 1998*; *Shmerling et al.,*

**eLife digest** Mammalian cells, including human cells, contain high levels of a protein called actin. This protein is essential for many of the processes that organisms use to develop and survive. For example, filaments of actin maintain the shape of cells, and help generate the forces needed for cells to move and divide.

As in many other animals, every cell in the human body contains two related actin proteins – known as β-actin and γ-actin. These proteins are made from almost identical amino acid building blocks. Yet the genes that encode these two proteins vary much more. The two actin proteins also play different roles: disrupting the gene for β-actin causes mouse embryos to die, but mice without the gene for γ-actin develop almost like normal. It was not fully understood how these almost identical proteins could perform such different roles.

Earlier studies exploring the mechanisms that underlie the unique roles of β- and γ-actin focused on the differences in their amino acid sequences. Now, Vedula, Kurosaka et al. test the hypothesis that the differing roles of these two actin proteins are due to the pronounced differences in the DNA sequences of their genes.

A gene-editing technique called CRISPR/Cas9 was used to make small changes to the mouse gene for β-actin so that it coded for the γ-actin protein. As a consequence, these mice did not make any β-actin protein and instead made the γ-actin protein from a mostly intact gene for β-actin. These mice lacking the β-actin protein survived as normal and were fertile. The shape of their organs and the movement of their cells – two other major processes that need β-actin – were also unaffected. Hence, the γ-actin protein can substitute for β-actin when the β-actin gene is intact.

These observations imply that it is the DNA sequence of the gene rather than the amino acid sequence of the protein that determines the essential role of β-actin in cell migration and the organism's survival. The next step will be to see if other proteins work in a similar way. If so, this mechanism might allow scientists to discover new ways to fine-tune how proteins behave in healthy and diseased human cells.

DOI: https://doi.org/10.7554/eLife.31661.002

2005; *Strathdee et al., 2008*; *Tondeleir et al., 2013*, *2014*), while γ-actin knockout in mice has a much milder phenotype that does not interfere with animal survival during embryogenesis (*Belyantseva et al., 2009*; *Bunnell and Ervasti, 2010*). This lethality of β-actin knockout in mice can be rescued by a targeted knock-in of the β-actin cDNA (*Tondeleir et al., 2012*), suggesting that the coding region of this gene is far more important for organism's survival than any non-coding elements affected by the β-actin knockout. The underlying mechanisms conferring such functional differences to these two nearly identical protein isoforms are unknown.

Several distinct features of the two actin isoforms have been proposed as the mechanism conferring unique roles to the two proteins. Some biochemical differences between β- and γ-actin were observed in polymerization assays both in vitro and in vivo (*Bergeron et al., 2010*; *Kapustina et al., 2016*; *Müller et al., 2013*). In addition, β-actin mRNA, unlike γ-actin mRNA, gets spatially targeted to the cell periphery via zipcode-mediated transport (*Hill and Gunning, 1993*; *Kislauskis et al., 1997*). Finally, β- and γ-actin actin can differentially regulate gene expression of a subset of cytoskeleton proteins (*Tondeleir et al., 2013*, *2014*). Despite these differences, no study to date has definitively identified the key functional determinants that confer unique functions to the mammalian actin isoforms in organism's survival, or determined whether these determinants reside at the amino acid level.

Here, we used targeted editing of the mouse genome to test the hypothesis that β- and γ-actin functions in vivo are defined by their nucleotide, rather than their amino acid sequence. Although previous studies have shown that disruption of the β-actin gene critically impacts embryonic development and organism survival (*Bunnell et al., 2011*; *Shawlot et al., 1998*; *Shmerling et al., 2005*; *Strathdee et al., 2008*; *Tondeleir et al., 2013*, *2014*), we demonstrate here that editing of the β-actin coding sequence to encode γ-actin protein without disruption of the rest of the β-actin gene does not affect mouse survival or produce a visible phenotype at the organismal or cellular level. This result shows that γ-actin protein is functionally capable of substituting for β-actin in the absence

of gene disruption. Thus, we demonstrate that the differences in in vivo functions of β- and γ-actin actin are ultimately determined by their nucleotide rather than amino acid sequence.

## Results

### β-actin nucleotide sequence, rather than amino acid sequence, defines its essential role in vivo

It has been previously found that knockout of β-actin gene in mice, unlike γ-actin, leads to early embryonic lethality – a result that definitively demonstrates its essential, non-redundant biological function (*Bunnell et al., 2011*). To test whether this essential function of β-actin is defined by its amino acid or nucleotide sequence, we used CRISPR/Cas9-mediated gene editing to introduce five point mutations into the native mouse β-actin gene (*Actb),* altering it to encode γ-actin protein without changing any of the features of the rest of the gene sequence ('beta-coded gamma actin', *Figure 1A*). We termed this edited gene *Actbc-g*, in which the native β-actin gene is nearly intact (with five point mutations within the first 10 codons) and contains the same promoter, as well as the same coding and non-coding elements, but the protein produced from this gene is identical to γ-actin. The outcome is no β-actin protein at all, enabling us to definitively test whether β-actin amino acid sequence, or its nucleotide sequence, is responsible for its essential function in organism's survival. If intact β-actin amino acid sequence is required, the mutant mice would die in embryogenesis, similarly to the β-actin knockout mice (*Bunnell et al., 2011*; *Shawlot et al., 1998*; *Shmerling et al., 2005*; *Strathdee et al., 2008*; *Tondeleir et al., 2013, 2014*). If the nucleotide sequence also contributes, these mice would be expected to survive longer than the β-actin knockout mice and have an overall milder phenotype. Finally, if the nucleotide sequence is the sole determinant of β-actin function, these mice would have no phenotype at all.

This gene editing strategy was successful, generating homozygous mouse mutants that contained no β-actin protein (*Figure 1—figure supplements 1–2*). Strikingly, *Actbc-g* mice appeared completely healthy, viable, and fertile with no signs of deficiencies previously seen in any of the β-actin knockout mouse models. These mice did not exhibit any visible defects in embryogenesis (*Figure 1B*), and appeared healthy and normal at birth and throughout life (*Figure 1C*) (observed until approximately 8 months old by the time of the submission of this study for publication). These mice also had normal fertility, as evidenced by litter sizes from *Actbc-g* homozygous breeding pairs that averaged 6.4 pups per litter (±0.38 SEM, n = 9), compared to the average litter size of 6.3 pups previously reported for their matching background wild type strain C57BL/6 (http://www.informatics.jax.org/silver/tables/table4-1.shtml). Thus, this result definitively proves that β-actin nucleotide sequence, rather than its amino acid sequence, determines the essential function of β-actin in vivo.

To test for possible milder defects in these mice, we analyzed the overall morphology and appearance of all major organs and body parts in newborn (P0) *Actbc-g* mice by sagittal sectioning and H&E staining, and found no overall differences or abnormalities between wild type and *Actbc-g* mice (*Figure 1D*), suggesting that the embryonic development in these mice occurs normally. Overall, *Actbc-g* mice appeared completely healthy and normal, suggesting that γ-actin protein encoded by the β-actin gene is fully able to functionally substitute β-actin's essential role in mouse survival and health.

To confirm the replacement of β-actin protein in these mice with the γ-actin protein, we performed quantitative western blots from several tissues where non-muscle actin isoforms are normally expressed at high levels, including brain, kidney, liver, and lungs (*Figure 2*). In all these tissues, loss of β-actin protein was accompanied by a prominent increase in γ-actin, without overall changes in total actin levels (*Figure 2* and *Figure 2—figure supplement 1*). Corresponding changes were also seen on 2D gels from these tissues, run under shallow pH gradient to separate actin isoforms (*Figure 2* and *Figure 2—figure supplement 2*).

We also performed a reciprocal experiment, using CRISPR/Cas9 gene editing to edit the mouse γ-actin gene to encode β-actin protein ('gamma-coded beta actin' or γc-β-actin, *Figure 2—figure supplements 3–4*). This strategy was only partially successful, resulting in replacement of the first three nucleotides to convert the N-terminal MEEE sequence of γ-actin into the MDDD sequence of β-actin, while failing to achieve I/V substitution at codon 10. However, given that the full deletion of γ-actin has a much milder phenotype than β-actin mouse knockout (*Belyantseva et al., 2009*;

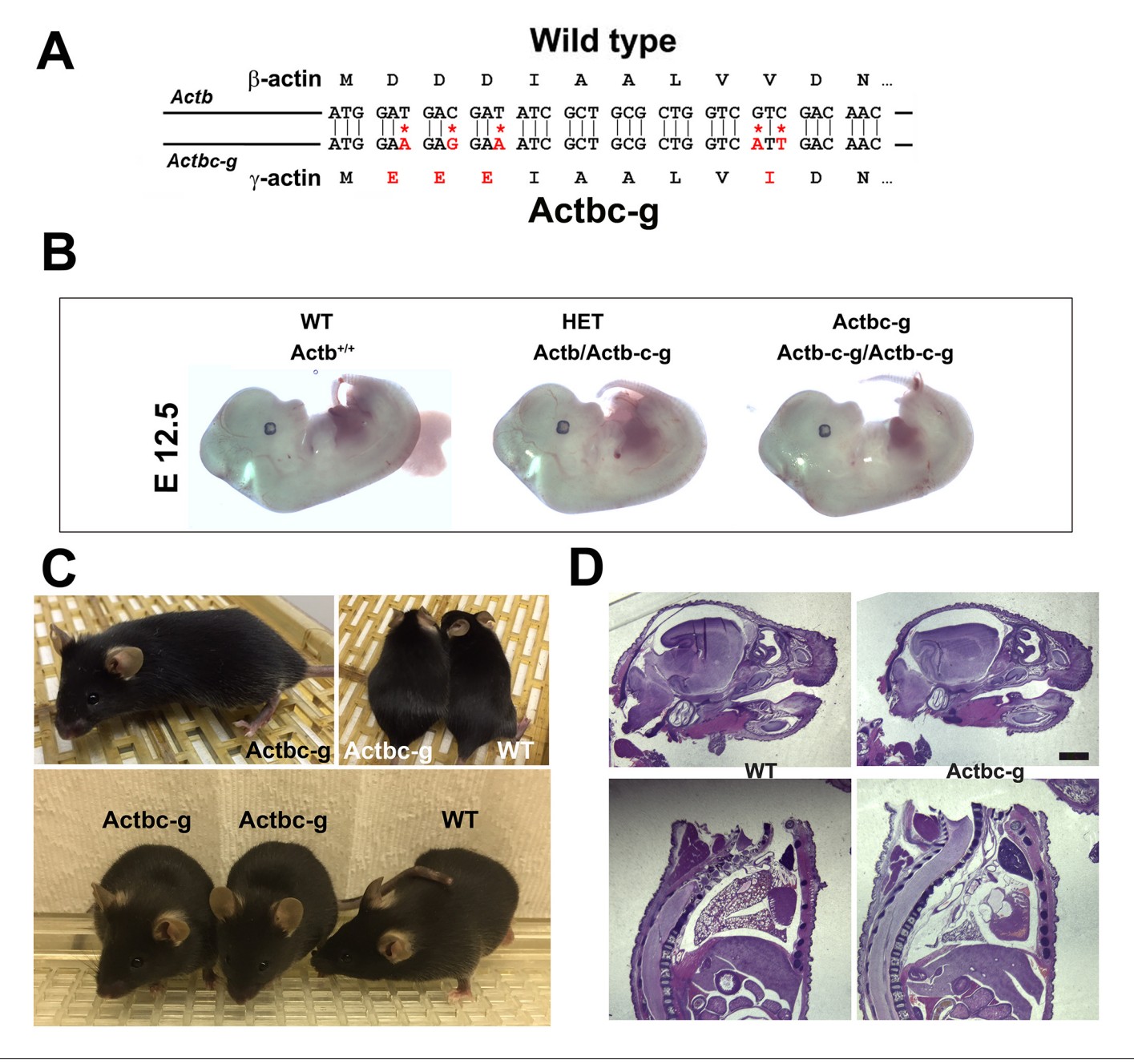

**Figure 1.** β -coded γ -actin (*Actbc-g*) mice exhibit no phenotypic changes compared to control. (**A**) CRISPR/Cas9 editing strategy used to generate *Actbc-g* mouse. (**B**) photos of *Actbc-g* E12.5 mouse embryos, with genotypes indicated. (**C**) photos of *Actbc-g* mice after gene editing, alone (top left) and next to age-matched (top right) and littermate wild type (WT) (bottom). Three mice from two different litters are shown. (**D**) H&E-stained sagittal sections of the heads (top) and bodies (bottom) of littermate P0 wild type (WT) and *Actbc-g* mice. Scale bar, 1 mm.

DOI: https://doi.org/10.7554/eLife.31661.003

The following figure supplements are available for figure 1:

**Figure supplement 1.** Generation of *Actbc-g* mouse.

DOI: https://doi.org/10.7554/eLife.31661.004

**Figure supplement 2.** Sequencing result for wild type *Actb* and the edited *Actbc-g* alleles.

DOI: https://doi.org/10.7554/eLife.31661.005

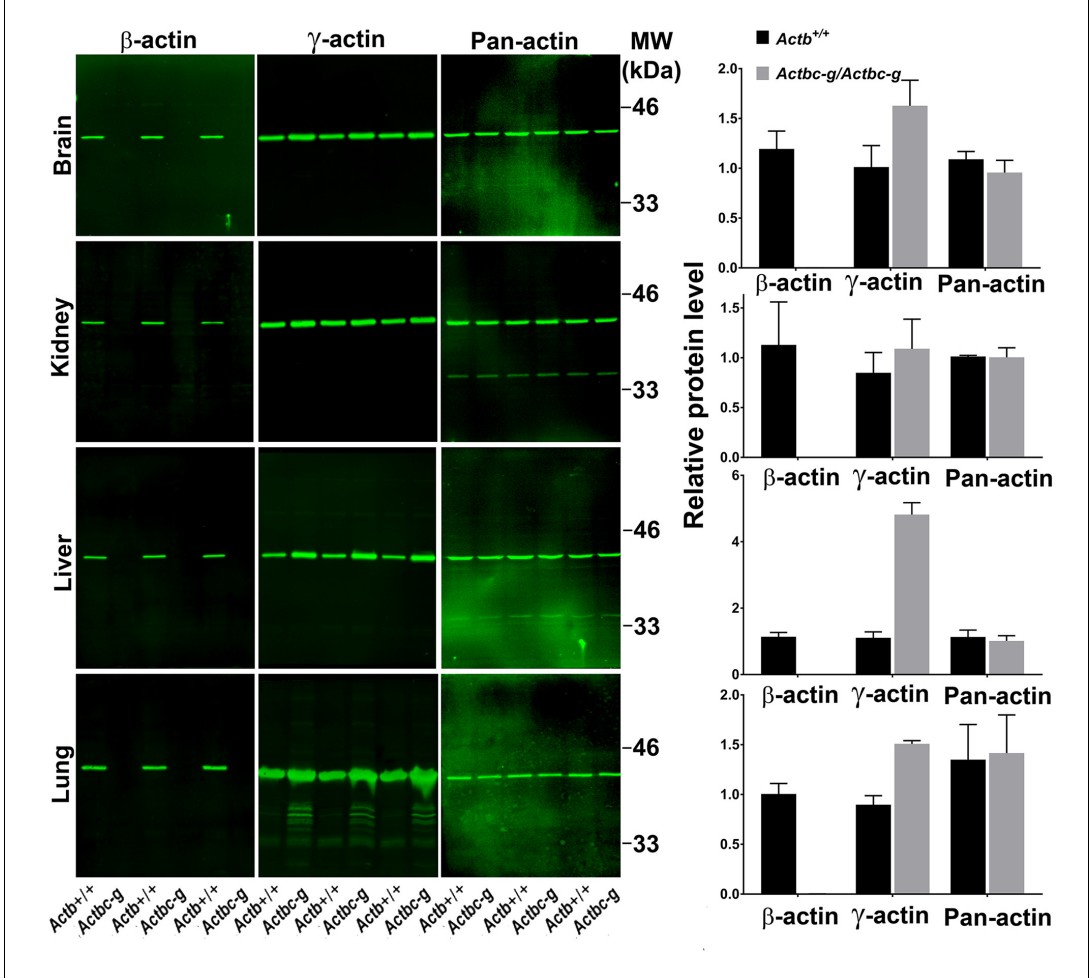

**Figure 2.** *Actb* gene editing abolishes β-actin protein from multiple organs and is accompanied by up-regulation of γ-actin without changing the total actin levels. Western blot analysis showing images (left) and quantifications (right) of whole tissue lysates from wild type (*Actb+/+*) and *Actbcg* mice. Fluorescence images obtained from the Odyssey gel imager are shown. For quantification, total fluorescence from the 43 kDa actin band was normalized to the loading control and to the actin level in the first lane for each blot. Error bars represent SEM, n = 3.

DOI: https://doi.org/10.7554/eLife.31661.006

The following figure supplements are available for figure 2:

**Figure supplement 1.** Actin levels in *Actbc-g* mice are similar to control.

DOI: https://doi.org/10.7554/eLife.31661.007

**Figure supplement 2.** 2D gel distribution of actin isoforms in wild type (top) and *Actbc-g* (bottom) mouse tissue lysates.

DOI: https://doi.org/10.7554/eLife.31661.008

**Figure supplement 3.** Generation of *Actg1c-b* mouse.

DOI: https://doi.org/10.7554/eLife.31661.009

**Figure supplement 4.** γ-coded β-actin (*Actg1c-b* mice exhibit no phenotypic changes compared to control.

DOI: https://doi.org/10.7554/eLife.31661.010

**Figure supplement 5.** Partial editing of the γ-actin gene to encode β-actin-like protein abolishes γ-actin protein from multiple organs.

DOI: https://doi.org/10.7554/eLife.31661.011

*Bunnell et al., 2011*; *Bunnell and Ervasti, 2010*), we did not pursue this further and analyzed the partially edited mouse instead. These *Actg1c-b* mice showed no detectable phenotype. At the same time, they showed disappearance of γ-actin protein and a corresponding increase in β-actin-like protein by western blots (*Figure 2—figure supplement 5*). These results suggest that γ-actin in vivo functions, like β-actin, is also defined by its nucleotide, rather than amino acid sequence.

## γ-actin protein expressed off β-actin gene supports normal cell migration

Since β-actin has been previously shown to play a major role in directional cell migration, and its knockout in cells leads to severe impairments in their actin cytoskeleton organization and their ability to migrate (*Bunnell et al., 2011*; *Tondeleir et al., 2012*), we next analyzed the actin cytoskeleton distribution and directional migration of mouse embryonic fibroblasts (MEF) derived from littermate wild type and *Actbc-g* mice. Despite the complete absence of β-actin protein in these cells, their actin cytoskeleton appeared similar to that of wild type cells. We detected no difference in F-actin levels in these cells (*Figure 3*), or in the morphology and appearance of the actin cytoskeleton (*Figure 3* and *Figure 4*).

To test the ability of these cells to migrate, we performed wound healing assays to measure the overall migration rates of the cell monolayers in wild type and Actbc-g. We also measured the directionality of single cell migration on fibronectin-coated dishes. In both assays, no difference was observed between the two cell types (*Figure 5* and *Figure 5—videos 1* and *2*), confirming that the actin isoform substitution did not result in any significant changes in these cells' ability to migrate. Thus, our data definitively demonstrate that the essential function of β-actin in vivo is defined by its nucleotide, and not its amino acid sequence.

## Actin isoform coding sequences show dramatic differences in ribosome densities that correlate with their essential biological function

We have previously reported that β- and γ-actin are differentially arginylated, due to differences in their mRNA nucleotide coding sequences (arising via silent substitutions), which leads to different translation dynamics, and the resulting rates of their accumulation in cells (*Buxbaum et al., 2014*; *Zhang et al., 2010*). To assess the potential differences in translation dynamics between the actin isoforms, we analyzed the global ribosome profiling data for these genes (mouse *Actb* and *Actg1*, encoding non-muscle β- and γ-actin, respectively) available at the GWIPS-Viz genome browser (http://gwips.ucc.ie; *Michel et al., 2014*) that aggregates the results of multiple ribosome profiling studies across genomes. Remarkably, composite data from 26 independently performed ribosome profiling studies from different mouse tissues show that the ribosome density on mouse β-actin mRNA is over a thousand fold higher than γ-actin (average ribosome density over the first 150 codons: 1351.607 for β-actin versus 1.289 for γ-actin, see also *Table 1* and *Supplementary file 1* and *2*) . These data suggest that the translation dynamics of β-actin in vivo is dramatically different from that of γ-actin. We next extended this analysis to the whole family of mouse actin genes and correlated the previously reported phenotypes resulting from mouse knockouts of different actin isoforms with the ribosome density number for each isoform (*Table 1*). This analysis shows that members of the actin family have vastly different ribosome densities, with β-actin being by far the highest, while γ-enteric smooth muscle actin is by far the lowest (ribosome density 0.377), suggesting that the intracellular accumulation rate for these two proteins, as well as their translation dynamics, should be vastly different from each other. In support, they also have different mRNA structures in the coding region (*Zhang et al., 2010*, *Supplementary file 3* and *4*). Notably, the actin isoforms that tend to become up-regulated in different knockout models are typically those with the closest ribosome density to the isoform that had been knocked out (*Table 1*). It appears likely that the success of functional compensation may also be linked to this number, directly or indirectly. For instance, alpha skeletal and alpha smooth muscle actins appear to partially cross-compensate for each other, and these isoforms are also the second and third highest by ribosome density, after β-actin. α-cardiac actin completely rescues knockout of α-skeletal actin, which has the closest ribosome density. In contrast, loss of α-cardiac actin cannot be substituted for by γ-enteric smooth muscle actin, which has a ~10 fold lower ribosome density. These results suggest the actin isoform with similar ribosome density can plausibly compensate for the loss of one of the isoforms. In agreement, given the orders of magnitude difference in ribosome density between β-actin and other actin isoforms, none of the other actin isoforms can compensate for the loss of β-actin. We propose that changes in ribosome density arising from silent substitutions in nucleotide sequence, affect translation dynamics and protein accumulation rates, which in turn regulate functional diversity of actins.

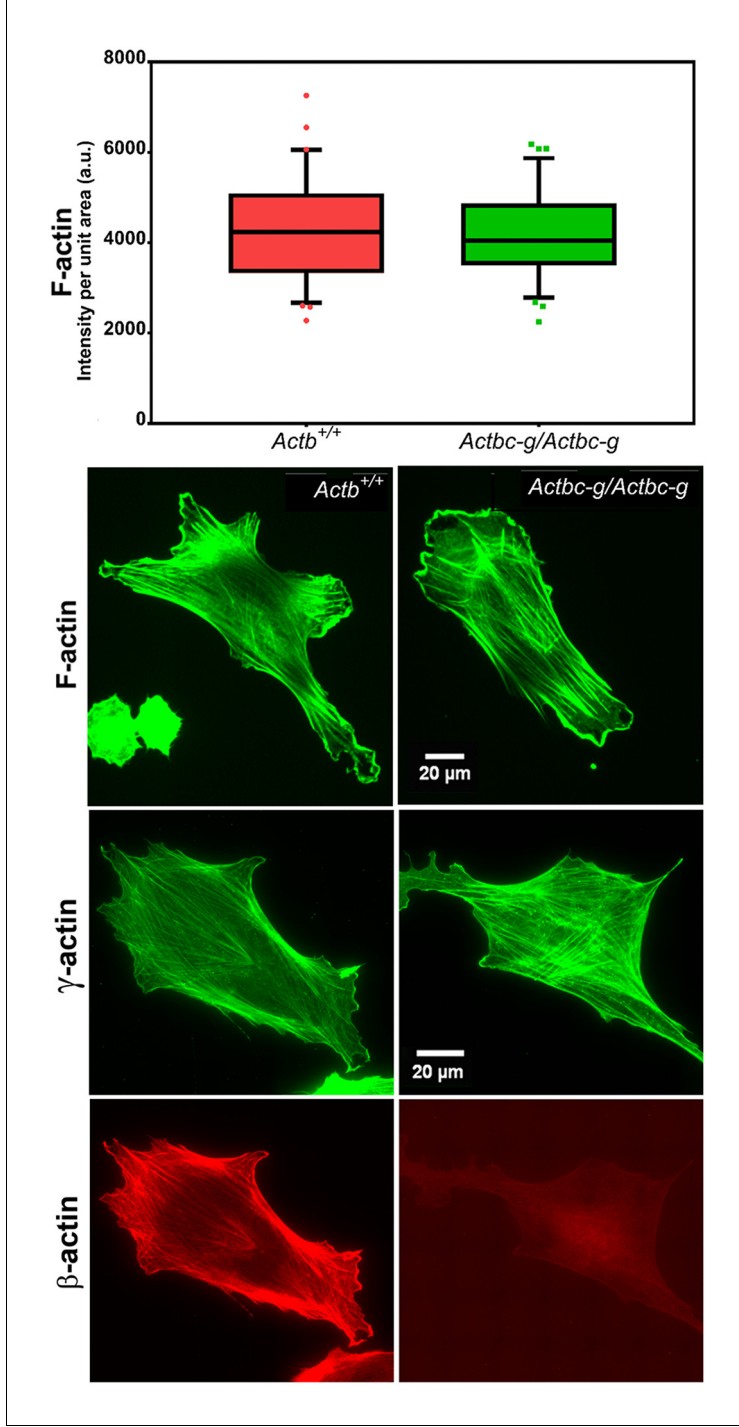

**Figure 3.** Mouse embryonic fibroblasts derived from *Actbc-g* mice have normal actin cytoskeleton, despite complete lack of β-actin. Top, quantification of total F-actin detected by Phalloidin-AlexaFluor488 staining of wild type (Actb$^{+/+}$) and *Actbc-g* primary mouse embryonic fibroblasts. Numbers were averaged from 69 cells in WT and 76 cells in *Actbc-g*, obtained from two different primary cultures independently derived from two different littermate embryos for each set. Bottom, representative images of both cell types stained with Phalloidin-AlexaFluor488 or antibodies to both actin isoforms as indicated.

DOI: https://doi.org/10.7554/eLife.31661.012

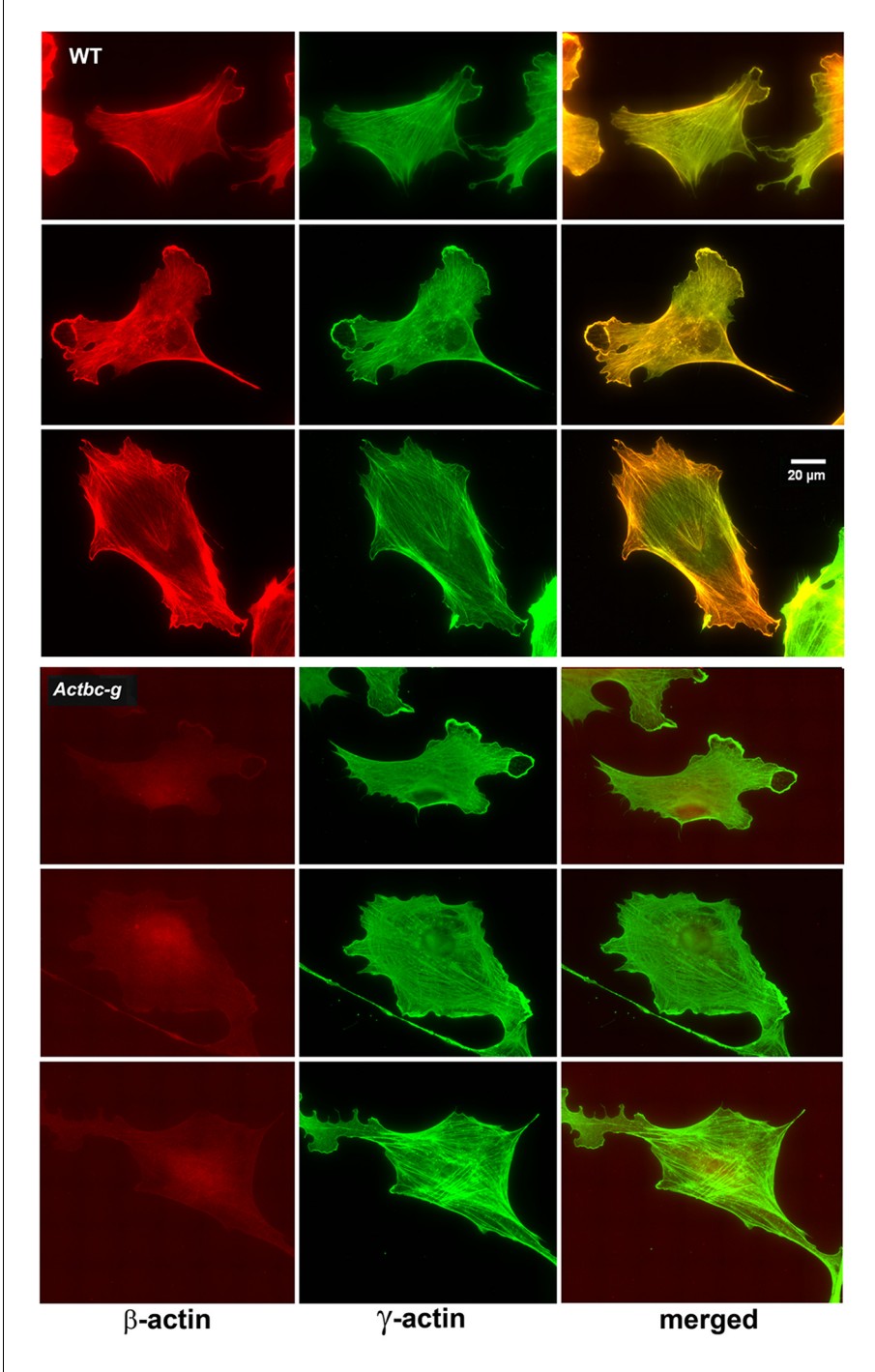

**Figure 4.** Mouse embryonic fibroblasts show no major changes in morphology and actin distribution. Representative images of wild type (WT) and *Actbc-g* primary mouse embryonic fibroblasts stained with antibodies to both actin isoforms as indicated.

DOI: https://doi.org/10.7554/eLife.31661.013

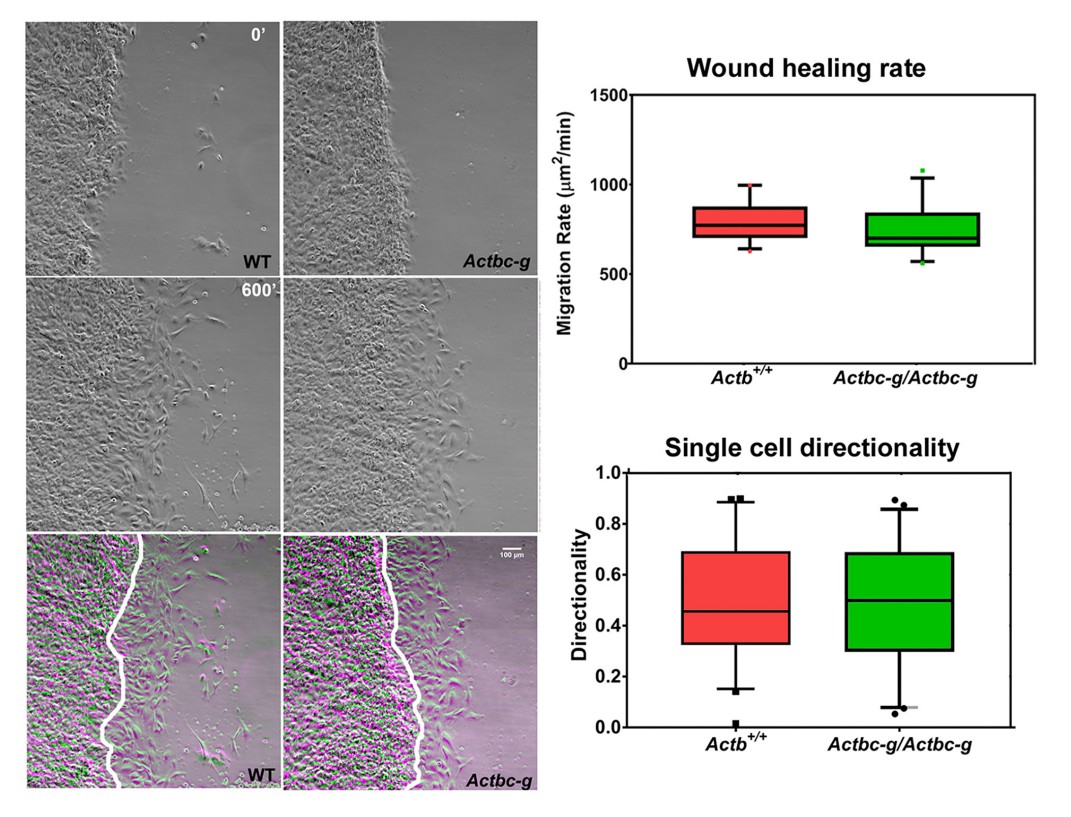

**Figure 5.** Mouse embryonic fibroblasts derived from *Actbc-g* mice migrate at normal rates. Left, phase contrast images of the first (0') and last (600') frame taken from a representative time lapse videos of the WT and *Actbc-g* cells at the edge of a monolayer migrating into an infinite scratch wound. Overlay of the two frames is shown in the bottom row. Scale bar, 100 μm. Right top, quantification of the cell migration rate as μm²/min, WT: n = 28, *Actbc-g*: n = 29 averaged from two independently derived primary cultures for each set. See *Supplemental Videos 1* and *2*. Right bottom, quantification of cell directionality in single cell migration assays (calculated as persistence over time, WT: n = 49, *Actbc-g*: n = 50) suggests that single cell migration was not affected in *Actbc-g* cells.

DOI: https://doi.org/10.7554/eLife.31661.014

The following videos are available for figure 5:

**Figure 5—video 1.** Time lapse video showing migration of control fibroblasts into the infinite scratch wound.

DOI: https://doi.org/10.7554/eLife.31661.015

**Figure 5—video 2.** Time lapse video showing migration of *Actbc-g* fibroblasts into the infinite scratch wound.

DOI: https://doi.org/10.7554/eLife.31661.016

## Silent substitutions in the coding sequence as a potential global regulator of homologous protein isoforms in eukaryotic genomes ('silent code')

We next used sequence analysis and Ribo-seq data profiling to test whether this type of silent substitution dependent protein regulation may potentially be applicable to other closely related protein isoform families in the mouse genome. First, to identify all protein families encoding highly similar proteins, we searched the mouse genome for proteins that are over 90% identical both in length and in their amino acid sequence over their entire length, and are encoded by different genes. This search yielded 741 families in mouse, encoding nearly 4000 different open reading frames. Next, we compared ribosome density numbers for different protein isoforms within these families using published ribosome profiling data found on GWIPS-Viz genome browser, and sorted these families by the normalized standard deviation (SD) of difference between ribosome densities for different members within the family. We discarded those families in which the ribosome density on the highest translating gene was less than 1, suggesting that their relative abundance in the polysome fraction is negligible (e.g., the majority of olfactory receptors). In the resulting list, we selected the top 100 families with SD of ribosome densities of 1.4 and above

**Table 1.** Severity of the actin isoform knockout phenotypes and their cross-compensation for each other correlate with their ribosome density.

See (*Perrin and Ervasti, 2010*) for the references on the isoform knockout data. * From heterozygotes (since homozygous knockout is embryonic lethal) and knockout MEFs. † Tissue specific upregulation of different isoforms See (*Bunnell and Ervasti, 2010*).

| Name | Gene symbol | NCBI accession number, protein | NCBI accession number, mRNA | Composite ribosome density | Mouse knockout phenotype | Other actin isoforms upregulated upon knockout |
|---|---|---|---|---|---|---|
| β-cytoplasmic actin | *Actb* | NP_031419 | NM_007393 | 1351.607 | Early embryonic lethality | *Acta2*; some *Actg1** |
| α-smooth muscle actin | *Acta2* | NP_031418 | NM_007392 | 53.781 | Viable, with vascular contractility and blood pressure defects | *Acta1* |
| α-skeletal actin | *Acta1* | NP_033736 | NM_009606 | 10.267 | Muscle weakness; postnatal lethality | *Acta2* and *Actc1* |
| α-cardiac actin | *Actc1* | NP_033738 | NM_009608 | 3.872 | Perinatal lethality | *Acta2* and *Acta1* |
| γ-cytoplasmic actin | *Actg1* | NP_033739 | NM_009609 | 1.289 | Viable, with growth defects and progressive deafness | *Acta2*, *Actb*, *Acta1*, and *Actc1*† |
| γ-enteric smooth muscle actin | *Actg2* | NP_033740 | NM_009610 | 0.377 | Unknown | Unknown |

DOI: https://doi.org/10.7554/eLife.31661.017

The following source data available for Table 1:

Source data 1. Composite ribosome profiling data for the actin gene family, plotted in logarithmic scale.
Bottom panel shows the coarse curves for the data on top.
DOI: https://doi.org/10.7554/eLife.31661.018

Source data 2. Ribosome profiling data for the individual members of the actin family, plotted in logarithmic scale.
DOI: https://doi.org/10.7554/eLife.31661.019

Source data 3. Predictions of the secondary structures for β− and γ− actin coding sequences.
Plot shows distance in nucleotides (x axis, 0 indicates the first ATG of the coding sequence) versus free energy (y axis).
DOI: https://doi.org/10.7554/eLife.31661.020

Source data 4 Predictions for the initial regions of the β− and γ− actin coding sequence, compared to their codon-switched versions.
Plot shows distance in nucleotides (x axis, 0 indicates the first ATG of the coding sequence) versus free energy (y axis). β-coded γ− actin mRNA is predicted to have a more relaxed structure around the translation initiation site, while being indistinguishable throughout the rest of the sequence.
DOI: https://doi.org/10.7554/eLife.31661.021

and classified them by functions (*Supplementary file 1*). We also performed the same analysis in two other vertebrate genomes, human and zebrafish (*Supplementary files 2* and *3*) and cross-referenced the identity of the top families to each other.

The top families identified in these searches were similar in all three vertebrate genomes, and contained the actin family as one of the top hits. The other uniformly present top hits included multiple classes of histones and tubulins. Notably, similar analysis of *D. melanogaster* and *C. elegans* genomes (*Supplementary files 4* and *5*) also yielded actins, tubulins, and histones as the top hits. Thus, the occurrence of families of nearly identical proteins with vastly different ribosome densities may be broadly common in higher eukaryotes, supporting our idea that this feature may underlie their functional regulation.

We propose that posttranscriptional regulation of highly homologous proteins in seemingly redundant protein families through silent nucleotide-encoded translation dynamics constitutes a novel type of functional regulation that governs some of the most essential functions in eukaryotic genomes. We propose to term this regulation 'silent code', to reflect its origin in silent substitutions within the genetic code.

## Discussion

Our results demonstrate a novel determinant – silent substitutions in nucleotide sequence – that drives the differential functions of non-muscle actin isoforms, a mechanism potentially applicable to other members of the actin family and other highly similar proteins in eukaryotic genomes. In the case of non-muscle actins, our finding resolves decades of controversial studies and reconciles a body of seemingly contradictory results obtained in the attempts to address functional distinction between β- and γ-actin (*Bergeron et al., 2010*; *Dugina et al., 2009*; *Kapustina et al., 2016*; *Patrinostro et al., 2017*). Our data demonstrate that the nucleotide sequence of the gene, rather

than the amino acid sequence of the encoded isoform, determines the absolute requirement of β-actin for organism's survival.

Closely related protein isoforms can exhibit functional differences which can be attributed to one or more of the following three sources. First, variations at the amino acid level can cause profound differences in protein function. Second, variations in mRNA properties due to differences in their coding sequence and UTR regions can strongly affect mRNA localization, stability, and translatability via secondary structure and codon usage. Finally, variations in gene intron sequences, promoter, and enhancer regions can contribute to the overall gene regulation, expression levels, and tissue specificity. Contribution of each of these levels to actin isoform function has been extensively investigated in prior studies. While β-actin and γ-actin isoforms share a remarkable conservation at the amino acid level, with just four homologous amino changes at their N-termini, these two proteins have been shown to have slightly different polymerization kinetics (*Bergeron et al., 2010*) and to differentially interact with cofilin (*Kapustina et al., 2016*) and non-muscle myosin isoforms (*Müller et al., 2013*). At the level of mRNA, actin 3'UTRs are isoform-specific and evolutionarily conserved, suggesting that they play important roles in vivo (*Erba et al., 1986*; *Hill and Gunning, 1993*). A plethora of literature elucidates the importance of β-actin 3'UTR for the localization of the transcript (see, e.g. (*Condeelis and Singer, 2005*) for a comprehensive review). It has also been shown that γ-actin mRNA induces a ribosome pausing event, resulting in its slower translation compared to β-actin, a mechanism that drives their differential arginylation (*Zhang et al., 2010*). An alternative poly A site in β-actin mRNA increases its translation levels (*Ghosh et al., 2008*). Finally, at the gene level, several studies point to roles of various regulatory elements in the actin isoforms. γ-actin gene contains a unique and highly conserved intron III (*Lloyd and Gunning, 1993*), and an alternatively spliced exon 3a (*Drummond and Friderici, 2013*). β-actin exhibits both 3'UTR-dependent (*Lloyd and Gunning, 1993*; *Lyubimova et al., 1999*) and 3'UTR-independent (*Lloyd et al., 1992*; *Schevzov et al., 1992*) feedback regulation of gene expression.

Our prior data showed that silent substitutions in β- and γ-actin coding sequence, amounting to a 13% overall difference, confer changes in the rates of their accumulation in the cell, and in their posttranslational modifications (*Zhang et al., 2010*). We show here, by analysis of publically deposited RiboSeq datasets, that β- and γ-actin mRNAs dramatically differ in ribosome densities, suggesting vastly different translation dynamics and likely different accumulation rates in cells. We propose that these silent substitutions in the nucleotide coding sequence may be key determinants that drive β- and γ-actin function. It is possible, however, that at the gene level additional non-coding elements and the untranslated regions (UTRs) of the actin mRNA may also contribute to the functional distinction between the actin isoforms. This possibility requires further investigation.

Ribosome profiling experiments show that among the 6 members of the actin family in mice, β-actin by far has the highest representation in the polysome fraction (*Ingolia et al., 2009*). We propose that this confers the cell with the ability to spatially and temporally fine tune translation of β-actin and the local and global rate of its intracellular accumulation (*Buxbaum et al., 2014*; *Ströhl et al., 2017*), and thus makes β-actin the most essential of the actin isoforms and cannot be compensated for when deleted. The nearly identical non-muscle γ-actin appears to be more functionally redundant and can be largely compensated for by other actins that have similar ribosome densities (*Perrin and Ervasti, 2010*), *Table 1*, and *Supplementary file 2* and *3*). Our findings suggest that actin isoforms with similar ribosome densities and translation dynamics are more likely to compensate for each other's functions, while β-actin, with the highest-ribosome-density is unique in its functional significance. Elucidating the exact contribution of translation dynamics to actin isoform functions constitutes an exciting direction of future studies.

Our result that targeted genome editing of mouse β-actin gene to encode for γ-actin protein leads to no apparent phenotype proves for the first time that the major determinants of β-actin's essential function in vivo are encoded at the nucleotide level. Some of the existing data point to the possibility that this effect is mediated primarily or exclusively via the coding region, rather than the promoter, the UTR, or the intron sequences. Indeed, deletion of exons 2 and 3 of the β-actin gene that include β-actin translation initiation site but do not encompass the promoter region, UTR, or the non-coding elements in the rest of the gene leads to embryonic lethality (*Bunnell et al., 2011*). Notably, in *Actb* knockout mice other actin isoforms are up-regulated to compensate for the total actin dosage, but this promoter-mediated up-regulation is insufficient to rescue the phenotype of early embryonic lethality. At the same time, targeted insertion of the human β-actin coding sequence

into this region rescues embryonic lethality in these mice, further supporting the idea that the coding sequence plays a key role (*Tondeleir et al., 2012*). Data from our group previously showed that coding sequence drives differences in actin's posttranslational modifications, one of the forms of functional actin regulation (*Zhang et al., 2010*). While these studies do not fully exclude a potential contribution of non-coding elements, especially those that may be located between exons 2 and 3 in the gene, they strongly support our hypothesis that the coding region is primarily responsible for the uniquely essential role of β-actin in vivo. Elucidating the underlying hierarchy of silent substitutions among other modes of regulation, would further our understating of the various levels of factors affecting the function of different actin isoforms.

Despite the fact that non-muscle actin isoform genes have evolutionarily diverged >100 million years ago, they have retained remarkable sequence conservation, far higher than what would be expected if the synonymous substitutions in their coding sequence were completely randomized (*Erba et al., 1986*). This is consistent with our idea that actin isoform coding sequence exists under additional evolutionary pressure, over and above the conservation of amino acid sequence. We propose that at least some of this pressure is aimed to maintain the divergent translation dynamics within the actin family, in order to drive their divergent functions.

Our sequence analysis and data mining suggest that this mechanism of homologous protein regulation through 'silent code' may be globally applicable to multiple protein families throughout eukaryotic genomes. Notably, one of the top candidate families for this regulation in addition to actin – tubulin – is also a major cytoskeletal protein regulated by multiple posttranslational modifications. Some members of the tubulin family conceivably may be regulated via silent substitutions in their coding sequence affecting translation dynamics, like shown previously for non-muscle actins. While systematic mouse knockout data for the alpha and β− tubulin isoform families is not available, based on the analogy with the actin isoforms, we predict that tubulin β-V (*Tubb5*) is likely the most essential of the β− tubulin genes, whose knockout likely cannot be substituted by up-regulation of any other β− tubulin. Notably, the two members of the γ− tubulin isoform family, *Tubg1* and *Tubg2*, have been knocked out in mice, and appear to follow the same trend: knockout of *Tubg1*, the tubulin isoform with higher ribosome density, is embryonically lethal, while knockout of *Tubg2* with the nearly 200 fold lower ribosome density is not (*Yuba-Kubo et al., 2005*). Further systematic analysis of knockouts of homologous isoforms would enable establishing the universality of the 'silent code'.

## Materials and methods

### CRISPR/Cas9 mutagenesis and genotyping

C57Bl/6 strain was used to generate the gene-edited mice. The donor females were super-ovulated using 5 IU of PMSG followed 48 hr later with 5 IU HCG, after which the females were mated immediately to C57Bl/6 studs. MEGAshort-script T7 transcription kit (Ambion Thermo Fisher Scientific, Waltham, MA) was used for in vitro transcription of small guide sgRNA (gctgcgctggtcgtcgacaaCGG, where CGG is the Protospacer Adjacent Motif, PAM) as per manufacturer's protocol. mMESSAGE mMACHINE T7 transcription kit (Ambion) was used to synthesize Cas9 mRNA. MEGAclear transcription clean up kit (Ambion Thermo Fisher Scientific) was used to purify the synthesized RNAs. About 20 hr post HCG, the CRISPR solution was injected into zygotes via pronuclear injection at a concertation of: Cas9 mRNA: 100 ng/µL, template DNA: 100 ng/µL, and gRNA: 50 ng/µL. The zygotes were further cultured overnight in KSOM media using a 5% CO2 incubator. All the embryos which successfully cleaved to the 2 cell stage were transferred into recipient females via oviduct transfer. A founder female that was mosaic for the mutation was derived and crossed with a wildtype male to derive heterozygotes. One male and female heterozygote from the F1 generation were crossed to produce F2 generation. Two separate litters from the F2 generation produced two wildtype females, five heterozygote males, one heterozygote female, and three homozygote males.

Template DNA sequence:

5′CGGCTGTTGGCGGCCCCGAGGTGACTATAGCCTTCTTTTGTGTCTTGATAGTAGTTCGCCA TGGAAGAGGAAATCGCTGCGCTGGTCATTGACAACGGCTCCGGCATGTGCAAAGCCGGC TTCGCGGGCGACGATGCTCCCCGGGCTGTA 3′

The *Actb* gene has an EcoRV site which gets destroyed upon gene editing (*Figure 1—figure supplement 1*). We utilized restriction digestion of a PCR product produced from the *Actb* gene to

determine the genotype of the resulting mice. While wild type (*Actb*$^{+/+}$) mice gave two bands: 600 bp and 300 bp upon EcoRV digestion, mice homozygous for the mutations (*Actbc-g/Actbc-g*) give a single band at 900 bp. PCR products from mice that are heterozygous for the mutation (*Actb/Actbc-g*) gave three bands upon EcoRV digestion (*Figure 1—figure supplement 1*). The results were further verified by sequencing the 5' end of the *Actb* gene. In order to verify that β-actin protein was no longer produced, tail samples were lysed and a western blot was carried out using isoform-specific antibodies: mouse anti-β-actin (Clone 4C2, EMD Millipore, Burlington, MA, and Clone AC15, Sigma Aldrich, Burlington, MA), mouse anti-γ-actin (Clone 2C3 EMD Millipore).

## Cell culture

Primary Mouse Embryonic Fibroblasts (MEFs) were isolated from the back area of freshly euthanized E12.5 mouse embryos by tissue disruption and cultured in DMEM (Gibco) supplemented with 10% FBS (Gibco).

## Cell migration assays and imaging

Cell migration was stimulated by making an infinite scratch wound. The cells were allowed to recover for a period of 2 hr before imaging. Migration rates were measured as the area covered by the edge of the wound in the field of view per unit time using Fiji (NIH, Bethesda, MD). For measurements of directionality in single cell migration assays, primary MEFs were cultured on glass bottom MatTek (Ashland, MA) dishes coated with 5 µg/ml Fibronectin at low cell densities. 2 hr after seeding, cells were imaged at 10 min intervals for 4 hr. Single cells were tracked using Metamorph Track Objects (Molecular Devices, Sunnyvale, CA) module. The obtained total displacement was divided by total distance of the track to obtain a directionality score shown in *Figure 5*. All images for these experiments were acquired on a Nikon Ti microscope with a 10X Phase objective and Andor iXon Ultra 888 EMCCD camera.

## Immunofluorescence

To quantitate the amount of actin polymer, cells were seeded on coverslips in six well plates at 20,000 cells/well overnight and fixed in 4% (w/v) PFA at room temperature for 30 min. Cells were then stained with Phalloidin conjugated to AlexaFluor 594 (Molecular Probes, Eugene, OR). Images were acquired using Andor iXon Ultra 888 EMCCD camera at 40X and the total intensity of phalloidin per cell was measured using Metamorph (Molecular Devices).

## Western blotting

Tissues from 2 month old *Actb*$^{+/+}$ and *Actbc-g/Actbc-g* mice were collected and flash frozen in liquid nitrogen. Brain, Kidney, Liver and Lung tissues were ground and weighed. The samples were lysed directly in 4x SDS sample buffer (1:4 w/v). Equal volumes of the lysates loaded for SDS-PAGE and 2D gel electrophoresis. Following transfer of the gels, the blots were dried and stained with LI-COR REVRT Total protein stain as per manufacturer's protocol. Images were obtained using an Odyssey scan bed in the 700 nm channel. The blots were then blocked and incubated with primary antibodies for mouse anti-β-actin (Clone 4C2, EMD Millipore), mouse anti-γ-actin (Clone 2C3, EMD Millipore), and rabbit anti-pan-actin (Cytoskeleton, Inc., Denver, CO). Secondary antibodies against mouse and rabbit conjugated to IRDye800 were used to probe the blots and images were acquired in the 800 nm channel using Odyssey scan bed. The total protein intensity was used to account for loading differences and the obtained signals were normalized to the first lane in the blot.

2D gel analysis was performed by Kendrick Laboratories, Inc. (Madison, WI) as described by (*Burgess-Cassler et al., 1989*; *O'Farrell, 1975*) using shallow pH gradient for the first dimension to separate actin isoforms (pH 4–6, 4–8).

## Sequence and ribosome profiling data analysis

NCBI RefSeq mouse genome v. 10 was used for this analysis. GWIPS read density profiles for mouse chromosomes were mapped to the NCBI RefSeq CDS annotations to produce the profiles for mouse mRNAs. Characteristic ribosome density was computed as the average density in the 5' 150 codons of the CDS.

The longest protein encoded in the same locus, as annotated in the genome, was collected from the same locus. Sequences were clustered at 90% identity over 90% of length using blastclust program (https://www.ncbi.nlm.nih.gov/pubmed/17993672).

## Acknowledgements

We thank Dr. John Pehrson for helpful discussions throughout this project and Drs. John Pehrson and Kei Miyamoto for critical reading of the manuscript. This work was supported by NIH grants GM104003 and GM117984.

## Additional information

### Funding

| Funder | Grant reference number | Author |
| --- | --- | --- |
| National Institutes of Health | GM104003 | Anna Kashina |
| National Institutes of Health | GM117984 | Anna Kashina |

The funders had no role in study design, data collection and interpretation, or the decision to submit the work for publication.

### Author contributions

Pavan Vedula, Formal analysis, Validation, Investigation, Visualization, Methodology, Writing—original draft; Satoshi Kurosaka, Conceptualization, Data curation, Formal analysis, Validation, Investigation, Visualization, Methodology; Nicolae Adrian Leu, Visualization, Methodology; Yuri I Wolf, Resources, Formal analysis, Validation; Svetlana A Shabalina, Resources, Software, Formal analysis; Junling Wang, Validation, Investigation; Stephanie Sterling, Methodology; Dawei W Dong, Formal analysis; Anna Kashina, Conceptualization, Data curation, Formal analysis, Supervision, Funding acquisition, Validation, Investigation, Visualization, Methodology, Writing—original draft, Project administration, Writing—review and editing

### Author ORCIDs

Pavan Vedula (iD) https://orcid.org/0000-0002-9914-0008
Satoshi Kurosaka (iD) http://orcid.org/0000-0002-4365-9003
Anna Kashina (iD) http://orcid.org/0000-0002-0243-6866

### Ethics

Animal experimentation: This study was performed in strict accordance with the recommendations in the Guide for the Care and Use of Laboratory Animals of the National Institutes of Health. All of the animals were handled according to approved institutional animal care and use committee (IACUC) protocols (#805204) of the University of Pennsylvania.

### Decision letter and Author response

Decision letter https://doi.org/10.7554/eLife.31661.029
Author response https://doi.org/10.7554/eLife.31661.030

## Additional files

### Supplementary files

• Supplementary file 1. Mouse families of homologous protein isoforms showing the highest differences in ribosome densities between the members of each family. Families are grouped by function, starting with the most abundant ones.
DOI: https://doi.org/10.7554/eLife.31661.022

• Supplementary file 2. Human families of homologous protein isoforms showing the highest differences in ribosome densities between the members of each family. Families are grouped by function, starting with the most abundant ones.
DOI: https://doi.org/10.7554/eLife.31661.023

• Supplementary file 3. Zebrafish families of homologous protein isoforms showing the highest differences in ribosome densities between the members of each family. Families are grouped by function, starting with the most abundant ones.
DOI: https://doi.org/10.7554/eLife.31661.024

• Supplementary file 4. Drosophila families of homologous protein isoforms showing the highest differences in ribosome densities between the members of each family. Families are grouped by function, starting with the most abundant ones.
DOI: https://doi.org/10.7554/eLife.31661.025

• Supplementary file 5. C.elegans families of homologous protein isoforms showing the highest differences in ribosome densities between the members of each family. Families are grouped by function, starting with the most abundant ones.
DOI: https://doi.org/10.7554/eLife.31661.026

• Transparent reporting form
DOI: https://doi.org/10.7554/eLife.31661.027

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
