## [Decision Letter]

Thank you for submitting your article "Diverse functions of closely homologous actin isoforms are defined by their nucleotide, rather than their amino acid sequence" for consideration by *eLife*. Your article has been favorably evaluated by Anna Akhmanova (Senior Editor) and three reviewers, one of whom, Pekka Lappalainen, is a member of our Board of Reviewing Editors. One of the other two reviewers, Peter Gunning, has also agreed to reveal his identity.

The reviewers have discussed the reviews with one another and the Reviewing Editor has drafted this decision to help you prepare a revised submission.

*Summary:*

The mammalian non-muscle actin isoforms, β- and γ-actin, are highly homologous, but yet they display different localization patterns and functions in cells. Moreover, β-actin deletion results in embryonic lethality and cannot be compensated by up-regulation of other actin isoforms. Recent studies by the Kashina lab revealed that only β-actin is arginylated in cells. Moreover, they demonstrated that differences in the nucleotide coding sequences, instead of amino acid sequences, between β- and γ-actin actin isoforms are responsible for differences in their arginylation.

Here, Vedula et al. generated knock-in mice to test whether the differences in the in vivo functions between β- and γ-actin arise from differences in their nucleotide, rather than their amino acid sequences. Importantly, their data demonstrate that editing β-actin gene to encode γ actin does not result in detectable defects in animal development and survival, or in migration and cytoskeletal organization of MEFs isolated from the mutant embryos. These and other data presented in the manuscript propose that in highly homologous protein isoforms, including β- and γ-actin and perhaps also histone and tubulin isoforms, the functional differences may arise from specific features of the nucleotide sequences that regulate translational dynamics, and hence affect post-translational modifications.

Majority of the experiments presented in the manuscript appear very convincing and of good technical quality. However, the manuscript could be significantly improved by better presentation of the data, by performing some additional control experiments, and by refocusing the manuscript to the most important findings.

*Essential revisions:*

1) The claim that γ-actin coded by the β-actin gene leads to no impact on organism survival needs some quantitative data, which can be derived from mouse breeding records. What is the comparative litter size of the different genotypes? While it would be unfair to ask for lifetime data, it would be helpful to have some data indicating the% survival out to 6 or 12 months. If such analysis cannot be performed, the authors should remove all conclusions concerning organism survival from the manuscript.

2) The part of manuscript focusing on editing of γ-actin gene to encode β-actin is not particularly informative. Not all amino acids were mutated (i.e. the protein is a β-γ actin hybrid) and the analysis of the phenotypes was superficial considering that the γ-actin mutant mice display only rather mild phenotypes. Thus, these data should be either deleted from the manuscript, or the authors should carry out a much more detailed phenotypic analysis of the mice.

3) The data on cell migration (since this is the process, where β-actin has been specifically implicated) could be solidified by further experiments. For example, Figure 3 shows a MEF from the β-c-γ, which appears to have well defined lamellipodia at opposite poles of the cell. In addition, the β-c-γ cell in Figure 3 shows what appears to be thicker lamellipodia. This suggests that tracking single cell motility might reveal that the β-c-γ cells show a reduced persistence of directional movement; a manifestation of disrupted polarity. Similarly, the images in Figure 4. suggest that there is much more individual cell movement in the wild-type compared to the β-c-γ cells, which appear to move much more as a 'community'.

4) The authors should extensively modify the 'Discussion' to better acknowledge previous biochemistry work on β- and γ-actins, as well as to consider alternative explanations for their results.

---

## [Author Response]

Essential revisions:

1) The claim that γ-actin coded by the β-actin gene leads to no impact on organism survival needs some quantitative data, which can be derived from mouse breeding records. What is the comparative litter size of the different genotypes?

The data on Actbc-g litter sizes from homozygous breedings was added to the paper. The average litter size for Actbc-g mice is 6.4 pups (averaged from 9 litters), which is very similar to the reported 6.3 average litter size for the background C57Bl6 wild type mouse strain used for CRISPR editing. Both numbers are now referenced in the manuscript.

While it would be unfair to ask for lifetime data, it would be helpful to have some data indicating the% survival out to 6 or 12 months. If such analysis cannot be performed, the authors should remove all conclusions concerning organism survival from the manuscript.

The current oldest live mouse in our colony has a birth date of March 13, 2017 and is approximately 8 months old. We have several others, mostly 3-7 months old (and some younger ones), but many of them were euthanized for experiments at different ages, precluding accurate assessment of their life spans. During the course of the study we did not at any point observe spontaneous lethality in these mice. All this information has been added to the revised manuscript. Overall, many studies in the field make conclusion about survival at the weaning stage, after which the phenotype is considered viable, even in those cases where high mortality rates are observed later on. For this reason, and given that we observe absolutely no survival issues in our mice at any analyzed stage up to 8 months, we believe that our conclusion about survival is merited in this case. However, we read through the text to make sure each statement about survival is justified and supported, and removed those that appeared unwarranted. We hope the reviewers agree, and we would be glad to make further specific changes as needed.

2) The part of manuscript focusing on editing of γ-actin gene to encode β-actin is not particularly informative. Not all amino acids were mutated (i.e. the protein is a β-γ actin hybrid) and the analysis of the phenotypes was superficial considering that the γ-actin mutant mice display only rather mild phenotypes. Thus, these data should be either deleted from the manuscript, or the authors should carry out a much more detailed phenotypic analysis of the mice.

We agree that the data on γ actin gene editing is far inferior to the one about the β actin gene, and fully accept this critique from the reviewers. However, we feel that the γ coded β-like mouse does provide limited additional information relevant to this study. For this reason, we took the liberty of keeping it in the paper in a greatly downplayed form. We removed the separate section describing this mouse, and added it as a supportive piece of evidence to the first Results section and to the supplemental figures (Figure 2—figure supplement 2–Figure 2—figure supplement 4). We hope the reviewers find this permissible, and we would be glad to remove these data altogether if needed.

3) The data on cell migration (since this is the process, where β-actin has been specifically implicated) could be solidified by further experiments. For example, Figure 3 shows a MEF from the β-c-γ, which appears to have well defined lamellipodia at opposite poles of the cell. In addition, the β-c-γ cell in Figure 3 shows what appears to be thicker lamellipodia. This suggests that tracking single cell motility might reveal that the β-c-γ cells show a reduced persistence of directional movement; a manifestation of disrupted polarity. Similarly, the images in Figure 4. suggest that there is much more individual cell movement in the wild-type compared to the β-c-γ cells, which appear to move much more as a 'community'.

We fully agree with the reviewers and are grateful for this suggestion. To address this comment we performed single cell migration study to measure directionality of primary mouse embryonic fibroblasts of both genotypes, measured as total displacement by total distance. The directionality of both cell types is similar, suggesting that no changes in the migration of these cells occur because of the mutation. We also added more cell images to the revised Figure 3 and Figure 4 to illustrate the variation in cell morphology that may have caused the initial impression about these differences. We hope this addresses the reviewer’s comment.

4) The authors should extensively modify the 'Discussion' to better acknowledge previous biochemistry work on β- and γ-actins, as well as to consider alternative explanations for their results.

The Discussion has been expanded to include the papers pointed out by the reviewers and systematically consider alternative explanations of our results in the context of the published literature in the field.